# Mining heterogeneous clinical notes by multi-modal latent topic model

Zhi Wen[1]☯, Pratheeksha Nair[1]☯, Chih-Ying Deng[2], Xing Han Lu[1], Edward Moseley[2], Naomi George[3], Charlotta Lindvall[2]*, Yue Li[1]*

**1** School of Computer Science and McGill Centre for Bioinformatics, McGill University, Montreal, Quebec, Canada, **2** Dana-Farber Cancer Institute, Boston, Massachusetts, United States of America, **3** Brigham and Women's Faulkner Hospital, Boston, Massachusetts, United States of America

☯ These authors contributed equally to this work.
* yueli@cs.mcgill.ca (YL); charlotta_lindvall@dfci.harvard.edu (CL)

**Data Availability Statement:** All relevant data are available on Github: https://github.com/li-lab-mcgill/heterogeneous_ehr.

**Funding:** a) YL is supported by Natural Sciences and Engineering Research Council (NSERC)

## Abstract

Latent knowledge can be extracted from the electronic notes that are recorded during patient encounters with the health system. Using these clinical notes to decipher a patient's underlying comorbidites, symptom burdens, and treatment courses is an ongoing challenge. Latent topic model as an efficient Bayesian method can be used to model each patient's clinical notes as "documents" and the words in the notes as "tokens". However, standard latent topic models assume that all of the notes follow the same topic distribution, regardless of the type of note or the domain expertise of the author (such as doctors or nurses). We propose a novel application of latent topic modeling, using multi-note topic model (MNTM) to jointly infer distinct topic distributions of notes of different types. We applied our model to clinical notes from the MIMIC-III dataset to infer distinct topic distributions over the physician and nursing note types. Based on manual assessments made by clinicians, we observed a significant improvement in topic interpretability using MNTM modeling over the baseline single-note topic models that ignore the note types. Moreover, our MNTM model led to a significantly higher prediction accuracy for prolonged mechanical ventilation and mortality using only the first 48 hours of patient data. By correlating the patients' topic mixture with hospital mortality and prolonged mechanical ventilation, we identified several diagnostic topics that are associated with poor outcomes. Because of its elegant and intuitive formation, we envision a broad application of our approach in mining multi-modality text-based healthcare information that goes beyond clinical notes. Code available at https://github.com/li-lab-mcgill/heterogeneous_ehr.

## Introduction

Multitudes of clinical notes are generated within the electronic health records (EHR) for each encounter between a patient and healthcare providers. These notes are written by clinical experts with specialized domain knowledge and include a plethora of rich information not otherwise captured within the EHR's laboratory, imaging, billing, and administrative

Discovery Grant (RGPIN-2019-0621), Fonds de recherche Nature et technologies (FRQNT) New Career (NC-268592), and Microsoft Research. b) The funders had no role in study design, data collection and analysis, decision to publish, or preparation of the manuscript. c) No author receives salary from any of the above funders.

**Competing interests:** Microsoft Research does not alter our adherence to PLOS ONE policies on sharing data and materials.

documentation. Importantly, there exist overlapping sub-domains of medical knowledge which depends on the particular expertise of the author. Due to distinct medical domain knowledge, different note types often involve different clinical vocabularies. In particular, clinical notes authored by physicians may differ considerably in terms of vocabulary and content compared to those notes authored by registered nurses. While Latent Dirichlet Allocation (LDA) [1] is a popular approach to extract meaningful topics from documents, it assumes that all of the documents follow the same topic distributions. We hypothesize that by modeling different note types each with a distinct discrete distribution using a multi-modal latent topic model, we can improve the interpretability of the latent topics learned from the notes and generate a more accurate risk stratification of patients.

To this end, we propose a multi-note topic model (MNTM) that jointly infers distinct latent topic distributions corresponding to each distinct note type. As a proof-of-concept, we use the clinical notes from the Medical Information Mart for Intensive Care III (MIMIC-III) data [2] for 17,000 patients in the intensive care unit (ICU). Our goal is to develop an early prediction model of the risk of prolonged mechanical ventilation (PMV) and in-hospital mortality among ICU patients based solely on clinical notes data accrued during the first 48 hours of their ICU admission. Early prediction was selected as the unit of analysis because of its high clinical relevance. PMV and in-hospital mortality were selected because they are the conventional outcomes for early prognostication in the critical care literature [3].

## Related methods

Our method of latent topic modeling is distinct from several previous methods [1, 4–6]. While previous investigators have employed latent topic models for mining clinical notes, to the best of our knowledge, none of these methods consider distinct note types differently. Chen et al. (2015) applied LDA directly to the EHR data without considering multi-modality [4]. Pivovarov et al. (2015) described a multi-modal LDA that infers topics by data types, where clinical note is one of the four data types (billing code, laboratory tests, clinical notes, and prescription) [5] but does not distinguish between note types. This model only works with a fixed set of data types. Li et al. (2020) described a multi-modal topic model called MixEHR to jointly infer distinct topic distributions for each data type while imputing non-missing at random laboratory test results [7]. While MixEHR can generalize to any arbitrary data type, it has not been applied to the current problem of multi-note-type modeling. Therefore, we consider our current approach as a novel application of the multi-modal topic model.

## Methods

### Multi-modal latent topic model

We propose a multi-modal latent topic model (Fig 1). Suppose there are $K$ latent disease topics. Each topic $k \in \{1, \ldots, K\}$ under note type $t \in \{1, \ldots, T\}$ represents a distribution over the vocabulary, which is a vector of unknown word frequency $\phi_k^{(t)} = [\phi_{wk}^{(t)}]_{W^{(t)}}$ for $W^{(t)}$ distinct words in the vocabulary. We assume that the topic-specific word frequency $\phi_k^{(t)}$ follows a Dirichlet distribution with unknown hyperparameter $\beta_{wt}$. For each patient $j \in \{1, \ldots, D\}$, the disease mixture membership $\theta_j$ is generated from the $K$-dimensional Dirichlet distribution $Dir(\alpha)$ with unknown asymmetric hyperparameters $\alpha_k$. To generate a note token $i$ for patient $j$, a latent topic $z_{ij}^{(t)}$ under data type $t$ is first drawn from a categorical distribution $\theta_j$. Then a clinical feature $x_{ij}^{(t)}$ is drawn from a categorical distribution with rate equal to $\phi_{z_{ij}^{(t)}}^{(t)}$.

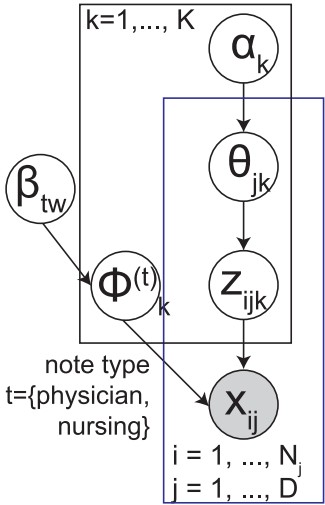

**Fig 1. Proposed multi-note latent topic model.**

Formally, we first generate global variables for the *K* topics:

$$\phi_k^{(t)} \sim Dir(\beta_t) : \frac{\Gamma(\sum_w \beta_{wt})}{\prod_w \Gamma(\beta_{wt})} \prod_w [\phi_{wk}^{(t)}]^{\beta_{wt}-1}$$

where *t* is the note types (e.g., $t \in$ {physician note, nursing note}). We then generate local variables for the patient topic mixture:

$$\theta_j \sim Dir(\alpha) : \frac{\Gamma(\sum_k \alpha_k)}{\prod_k \Gamma(\alpha_k)} \prod_k \theta_{jk}^{\alpha_k-1}$$

Given the topic mixture, we sample a topic for each token in note type *t* of each patient's note:

$$z_{ij}^{(t)} \sim Cat(\theta_j) : \prod_k \theta_{jk}^{[z_{ij}^{(t)}=k]}$$

We then sample a word for token *i* from topic distribution under topic $z_{ij}$:

$$x_{ij}^{(t)} \sim Cat(\phi_k^{(t)}) : \prod_w (\phi_{kw}^{(t)})^{[x_{ij}^{(t)}=w]}$$

Notably, the topic mixture $\theta_j$ is shared across note types and can therefore facilitate "borrowing" information between different note types when learning the topic distribution $\phi^{(t)}$.

To learn the model, we implemented a collapsed variational Bayesian algorithm [8]. Briefly, we first integrate out the Dirichlet variables because they are conjugate to the multinomial distribution of the tokens making the resulting inference much more efficient. We then approximate the expectations by first deriving the conditional distribution for the topic assignments $z_{ij}^{(t)}$ and then approximating their sufficient statistics by the variational parameters:

$$\gamma_{ijk}^{(t)} \propto \left(\alpha_k + \tilde{n}_{.jk}^{-(i,j)}\right) \left(\frac{\beta_{tx_{ij}^{(t)}} + [\tilde{n}_{x_{ij}^{(t)}.k}^{(t)}]^{-(i,j)}}{\sum_w \beta_{wt} + [\tilde{n}_{w.k}^{(t)}]^{-(i,j)}}\right) \tag{1}$$

where the notation $n^{-(i,j)}$ indicates the exclusion of token $i$ in patient $j$'s clinical note and the sufficient statistics are

$$\tilde{n}_{.jk}^{-(i,j)} = \sum_{t=1}^{T} \sum_{i' \neq i}^{N_j^{(t)}} \gamma_{i'jk} \tag{2}$$

$$[\tilde{n}_{w_t.k}^{(t)}]^{-(i,j)} = \sum_{j'=1}^{D} \sum_{i'=1}^{N_{j'}} [x_{i'j'}^{(t)} = w_t] \gamma_{w_t j'k}^{(t)} - [x_{ij}^{(t)} = w_t] \gamma_{w_t jk}^{(t)} \tag{3}$$

The learning algorithm therefore follows a variational Bayes expectation-maximization algorithm: E-step infers $\gamma_{ijk}^{(t)}$'s with Eq (1); M-step updates sufficient statistics $\tilde{n}_{.jk}$ and $\tilde{n}_{w.k}^{(t)}$ with Eqs (2) and (3), respectively. The EM update guarantees maximizing the evidence lower bound (ELBO) of the model under the mean-field variational distribution for independent topic assignments (i.e., $q(\mathbf{z}) = \prod_{t,i,j} q(z_{ij}^{(t)} | \gamma_{ij}^{(t)})$) [8].

Upon convergence of ELBO, we infer the respective variational expectations of the patient topic mixture and topics distribution:

$$\hat{\theta}_{jk} = \frac{\alpha_k + \tilde{n}_{jk}^{(.)}}{\sum_{k'} \alpha_{k'} + \tilde{n}_{jk'}^{(.)}}, \quad \hat{\phi}_{wk}^{(t)} = \frac{\beta_{wt} + \tilde{n}_{wk}^{(t)}}{\sum_{w'} \beta_{w't} + \tilde{n}_{w'k}^{(t)}}$$

Furthermore, we update the hyper-parameters by maximizing the marginal likelihood under the variational expectations via empirical Bayes fixed-point update [9, 10]:

$$\alpha_k^* \leftarrow \frac{a_\alpha - 1 + \alpha_k \sum_j \Psi(\alpha_k + \tilde{n}_{jk} + \tilde{m}_{jk}) - \Psi(\alpha_k)}{b_\alpha + \sum_j \Psi(\tilde{n}_{jk} + \sum_k \alpha_k) - \Psi(\sum_k \alpha_k)} \tag{4}$$

$$\beta_{wt}^* \leftarrow \frac{a_\beta - 1 + \beta_{wt} \sum_k \sum_w \Psi(\beta_{wt} + n_{w.k}^{(t)}) - KW_t \Psi(\beta_{wt})}{b_\beta + \sum_k \Psi(W_t \beta_{wt} + \sum_w n_{w.k}^{(t)}) - K\Psi(W_t \beta_{wt})} \tag{5}$$

where $\Psi(.)$ is the digamma function, $W_t$ is the vocabulary size under clinical note type $t$, the Gamma parameters are set to fixed values mainly for numerical stability: $a_\alpha = 1; b_\alpha = 0, a_\beta = 1, b_\beta = 100$.

## MIMIC-III note processing

From the entire cohort (all patients admitted to the ICU), we selected a subset, which we have called day-2 cohort. This subset includes the notes of patients that have been mechanically ventilated for at least two consecutive days. We used the entire cohort excluding the day-2 cohort, to train our unsupervised topic model and then used this trained topic model to infer topic mixtures of notes in the day-2 cohort, which are used for mechanical ventilation prediction.

For both cohorts, we performed a standard text preprocessing procedure including converting letters to lower case, removing punctuation, white spaces, stop words provided by Natural Language Toolkit library (https://www.nltk.org/), and words that appeared in fewer than 5 notes or in more than 15% of notes. After the preprocessing, each note had around 300 words on average. The vocabulary for physicians' notes contained 8948 words and the vocabulary for nursing notes contained 8076 words. In our study, the notes of an admission, instead of a patient, were grouped together as one document, and were therefore assumed to have one topic composition. While notes written in different admissions might have different focuses

on the topics, it is reasonable to assume notes within a single admission have mostly the same topics, including notes written by different professionals.

For the single-note-type model, we processed the notes in two different ways: (1) the same words from the different types were assigned the same word ID and their frequencies were the overall total sum over all types of notes (referred to as "single-note-type (same word)"); (2) the same words from different types were assigned different word IDs and their frequencies were computed separately (referred to "single-note-type (diff. word)"). For example, the word 'heartbeat' may occur in both a physician's note as well as a nursing note but is represented separately (as 'physician-heartbeat' and 'nurse-heartbeat'). For the proposed multi-note model, we differentiated such words by assigning different note types to them.

We evaluated our model's predictive performance by 5-fold cross-validation. Prolonged mechanical ventilation was defined as $\geq 7$ days because this time period represents a major clinical decision branch in a patient's care [11–13].

## Qualitative evaluation

We performed a qualitative evaluation of the topic cohesiveness. Topic cohesiveness was defined a priori as "relatedness of each term within the topic to a central disease process or health state". Cohesiveness was measured by a blinded physician using a 5-point scale. A second blinded physician with content expertise in critical care medicine reviewed the word clouds of each model in aggregate and provided a determination of the relative cohesiveness of the two models.

## Results

### Multi-note model improves PMV and mortality prediction

In each validation fold, we trained both the single-note models (single-note-type (same words) and single-note-type (diff. words)) and the multi-note model on the training set followed by a logistic regression model to predict the binary outcome of PMV also on the same training set. We used 50 topics for each of the 3 topic models. We experimented with 10, 30, 50 and 100 topics by measuring the perplexity on held-out documents and chose the best number of topics going forward.

We then predicted the PMV binary outcome on the validation set (Fig 2). We observed consistent improvement in terms of area under the receiver operating characteristic (ROC) curves (AUROC: 66.8% for multi note type, 66.0% for single note type (diff. words), 60.7% for single note type (same words)) and area under the precision-recall curve (AUPRC: 40.8% for multi note type, 39.2% for single-note-type (diff. words), 33.9% for single note type (same words)). In particular, the multi-type model achieved AUROC equal to 0.668 with standard deviation (std) equal to 0.008. Hence, the 95% confidence interval (CI) was $[0.668 - 1.96 \times 0.008/\sqrt{10}, 0.668 + 1.96 \times 0.008/\sqrt{10}] = [0.6630, 0.6730]$. The best single-note type model (diff-word) achieved on average $0.660 \pm 0.008$ std (i.e., [0.6550, 0.6650] 95% CI). Therefore, the AUROC of the multi-note model was higher than the best single-note model but the difference was not statistically significant at 95% CI. However, AUROC tends to be insensitive to unbalanced data. We therefore turned to AUPRC. In terms of AUPRC, the multi-note model achieved on average $0.408 \pm 0.007$ (std), while the best single-note model achieved on average $0.392 \pm 0.008$ (std), and the 95% confidence interval in terms of AUPRC were [0.404, 0.412] and [0.387, 0.397], respectively. This showed that the AUPRC of the multi-note model was significantly higher than the AUPRC of the best single-note model at 95% CI.

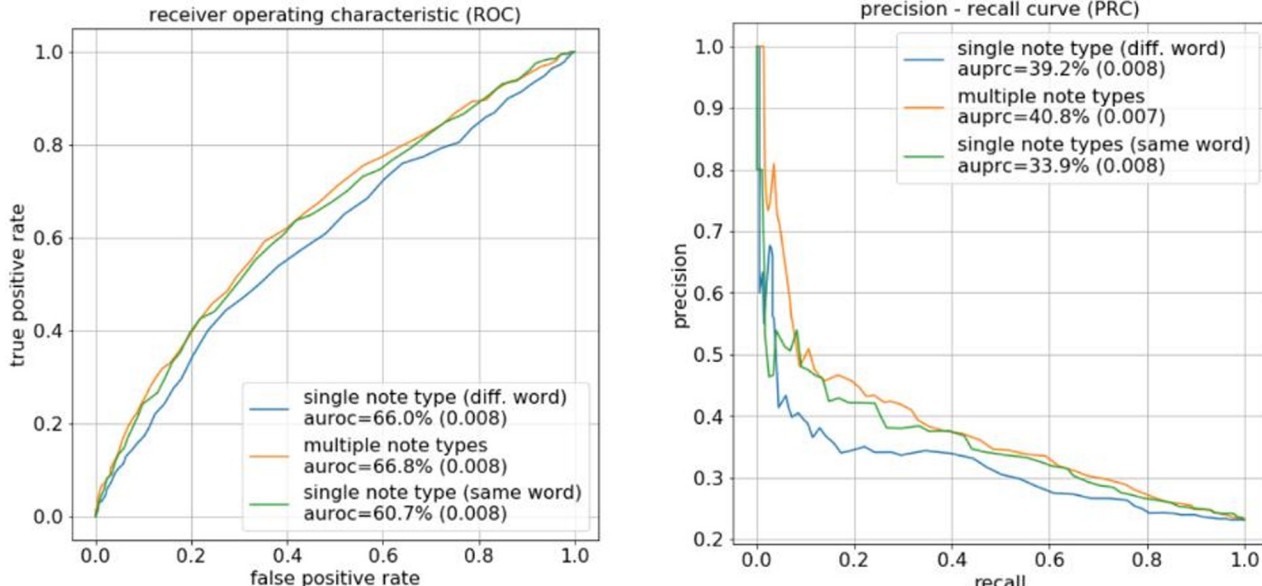

**Fig 2. ROC and precision-recall curve for binary PMV prediction.** We trained the two single-note topic models and the multi-note topic models on the first 48 hours of the clinical notes for each patient. We then trained a separate logistic regression classifier that took the patient-note topic mixture as input and predicted whether the patient is going to stay on MV for more than 7 days. The trained topic models and logistic classifiers were then applied to the test patients to make the predictions of PMV duration. The prediction accuracy was evaluated by ROC and precision-recall curves. The figure inset shows the AUROC and AUPRC values for each model, and the standard deviations across 10 random splits are in parenthesis.

To further illustrate the benefits of modeling multi-note types, we applied our approach to mortality prediction. Here we used the first 48 hours nursing and physician notes to predict in-hospital mortality. Same as the PMV application, we trained a 50-topic model for each approach and used the topic mixture memberships as an input to a logistic regression classifier for predicting mortality. We performed 5-fold CV to evaluate each method. In particular, each fold including 1560 admissions for evaluation and the remaining 4 folds including 6233 admissions total were used for training each topic model. We found that the multi-note model performed slightly better compared to single-note models, as measured by AUROC and AUPRC (S2 Fig in S1 File). On mortality prediction, the multi-note model achieved on average 0.861 ± 0.004 (std) in terms of AUROC and [0.859, 0.863] 95% CI. The best single-note (same-word) model achieved on average 0.845 ± 0.004 (std) and [0.843, 0.847] 95% CI. In terms of AUPRC, the multi-note model achieved on average 0.419 ± 0.011 (std) and [0.412, 0.426] 95% CI, while the single-note model achieved on average 0.404 ± 0.008 (std) and [0.399, 0.409] 95% CI. These indicated that both the AUPRC and AUROC of the multi-note model are significantly higher than those of the best single-note model at 95% CI.

By construction, the single type (diff. word) model operates over a vocabulary that is roughly twice as big as that of the single type (same word) model (because the same word coming from the two note types is treated as two different words). On the other hand, the multi type model operates on the same vocabulary as single type (same word), but counts the same word coming from different notes types differently. Therefore, to compare more fairly by controlling the impact brought by the effective "vocabulary size" (unique words that are seen by the models), we focused our subsequent analysis on the comparison between the multi-note type model and single-note-type (diff. words) model. For ease of reference, we rename the single-note-type (diff. words) model simply as single-note. We focus our analysis on PMV henceforth as it is less explored than mortality.

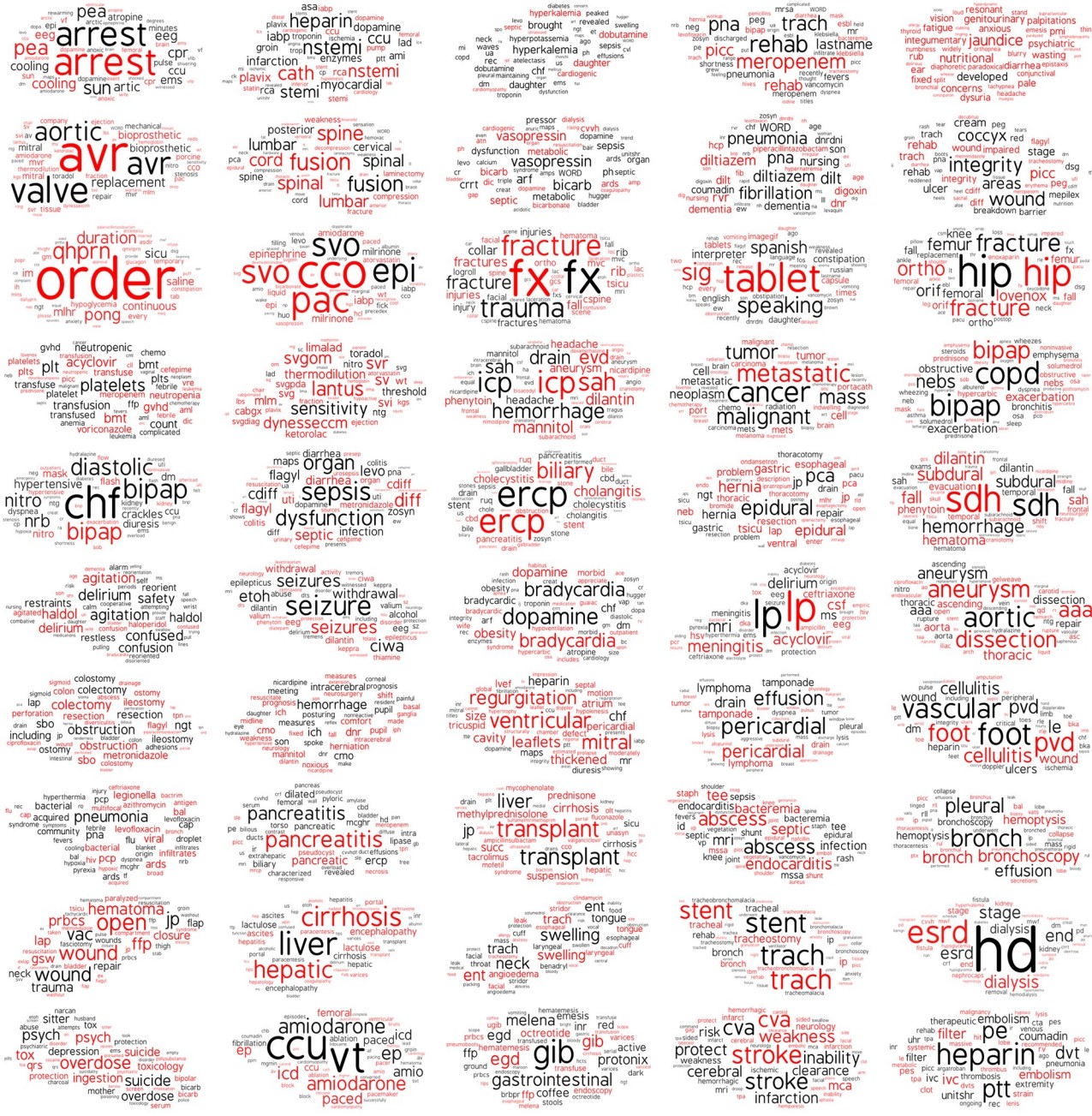

**Fig 3. Word clouds of the 50 topics from multi-note model.** Red indicates the words written by physicians and black indicates the words written by nurses.

### Evaluating the topic interpretability of single-note and multi-note topic models

To evaluate the interpretability of the single-note versus the multi-note topic model, we generated a word cloud representing each of the 50 topics in both models (i.e. 100 word clouds). Each topic's word cloud was comprised of the top 100 words within the topic, based on the inferred word probabilities under each topic. (S1 Fig in S1 File and Fig 3).

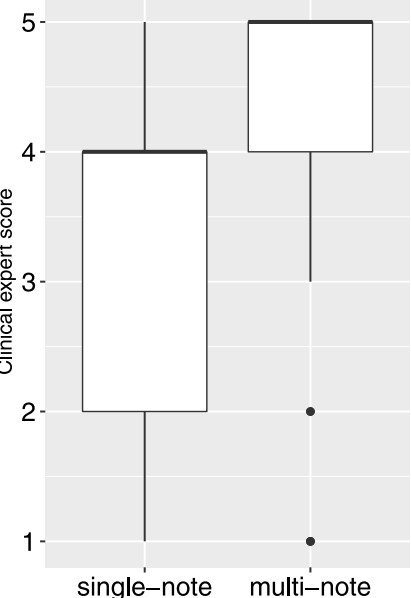

**Fig 4. Topic scores over 50 latent topics inferred by single-note and multi-note latent topic models from the 17,000 clinical notes.** The horizontal lines in the box represent the median and the box represent the range between 25% and 75% quartile of the data.

For the single-note model, the most common topic themes were "mixed" topics followed by topics pertaining to cardiology, gastroenterology, neurology and respiratory issues. The most common topic themes for multiple-note model were those pertaining to cardiology, gastroenterology, respiratory and neurology. The topics generated by the multi-note model had significantly more cohesiveness than the topics generated by the single-note model. In the multi-note model, most word clouds were comprised of words, phrases, or abbreviations that tracked closely with that topic's theme. By comparison, the topics extracted in the single-note model contained a greater amount of noisy, unrelated words. For example, the single-note model generated a topic themed "hematological" in which `pillow` was the most common word, and a topic themed "stroke" in which `adenoca` was a common word

In addition, we sought an unbiased quantitative evaluation of the topic interpretability. We asked a physician to manually review the general medical cohesiveness of each word-cloud in the single-note and multi-note model and rated from 1 (poor; irrelevant) to 5 (excellent; sticks to one common disease topic).

Quantitatively, the average interpretability score is 3.46 (± 1.15 standard deviation (std)) for single-note model and 4.22 (± 1.15 std) for multi-note model (Fig 4). We conducted a two-sided t-test between the physician ratings of the multi-note topic model and the single-note topic model (i.e., the standard LDA model) in R and obtained a p-value equal to 0.001298. This indicated that the difference between these two models in terms of physician's ratings is statistically significant. This trend was further confirmed by the content expert reviewer. The detail of the topic disease and cohesiveness score are listed in S1 and S2 Tables in S1 File.

## Correlating topics with mechanical ventilation duration

To gain further insights from the 50 learned topics, we inferred the 50-topic patient mixture memberships using the trained topic model. We then correlated the patient 50-topic mixture with the patient's total mechanical ventilation (MV) duration using only those patients' notes

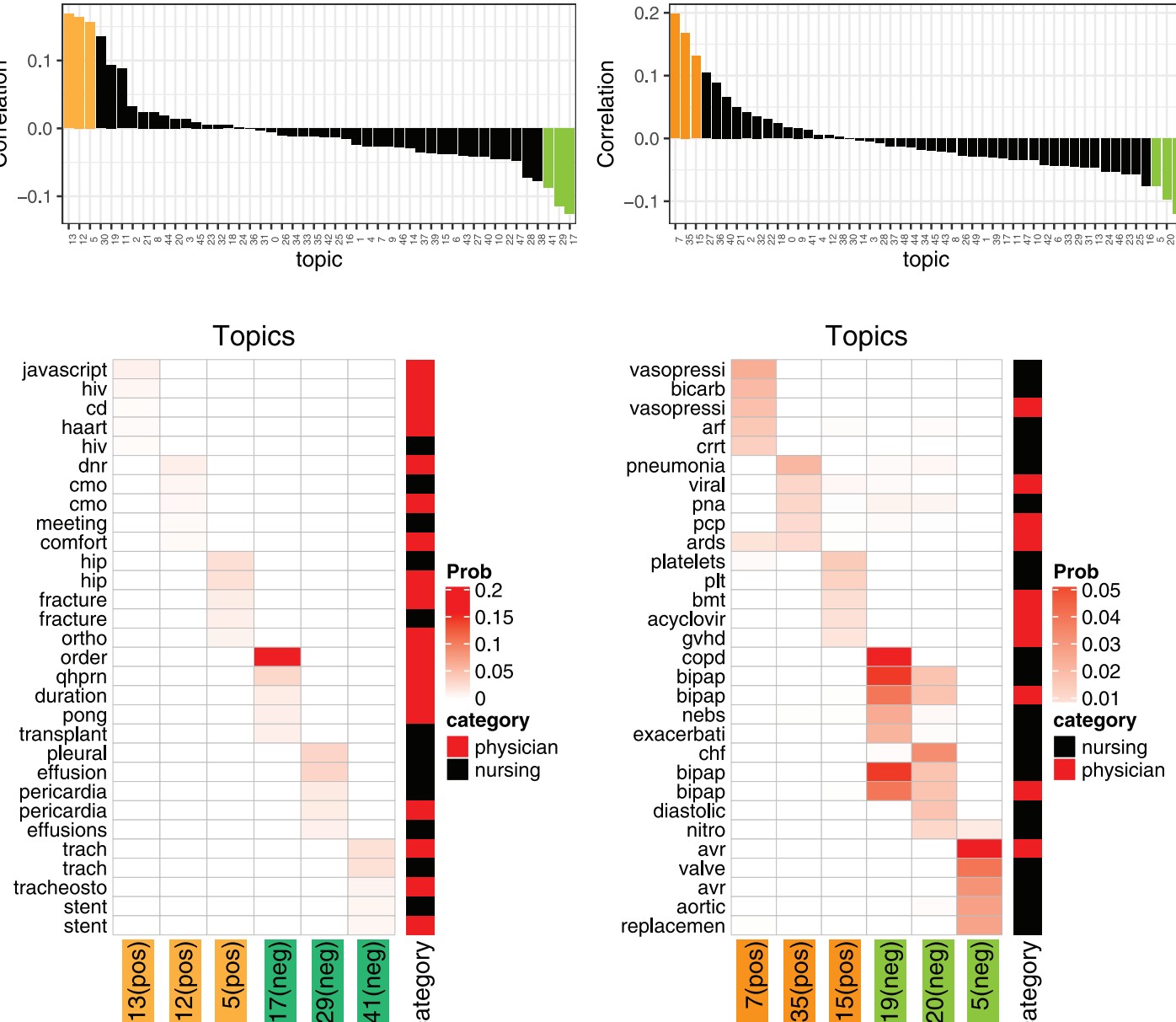

**Fig 5. Topic correlations with the MV duration.** We correlated the topic mixture with the MV duration and displayed the correlation as barplots We then visualize the top words for the the topics that are most positively and negatively correlated with PMV using **(a)** the single-note model; **(b)** the multi-note model. Only the 3 most positively and the 3 most negative correlated topics were shown for each model.

that were recorded within 48 hours of the their ICU admission (Fig 5 top panels). We chose Pearson's correlation coefficient because it is a normalized metric whose magnitude reflects the strength of linear correlation, in the range of -1 to 1, and the range restriction of the variables has no impact on the correlation. We also tried Spearman's and Kendall's correlation coefficients and observed similar results. We visualized the top 3 most positively correlated topics and the top 3 most negatively correlated topics for single-note model and multi-note model (Fig 5 bottom panels). The multi-note model clearly revealed more meaningful topics

related to MV duration. For example, the most correlated topics for MV from multi-note model was associated with septic shock followed by pneumonia. In contrast, the most correlated topic for MV from the single-note model is associated with 'javascript system error' along with some discrete and irrelevant terms and concepts. The most common negatively correlated topic for MV was chronic obstructive pulmonary disease (COPD) with acute exacerbation (AE) from the multi-note model and liver transplant with some sparse and unrelated terms from the single-note model.

## Discussion

Different types of medical specialists, such as physicians and nurses, hold distinct domains of medical knowledge. These differences are reflected in the language and terms that populate clinical notes. Existing methods of LTM treat notes authored by different types of medical specialists as the same by assuming all notes follow a homogeneous topic distribution. To the best of our knowledge, we are the first group to propose a model that applies separate analysis depending on the author type of the notes. Our simple and elegant multi-modal topic model showed the advantage of inferring distinct distributions of latent topics between physician and nursing notes. We demonstrated that the proposed multi-note model extracts more meaningful topics and improves the interpretability of the knowledge learned from the notes as compared to the single-note model. We also showed that our model confers, slightly but statistically significantly, more accurate prediction of duration of MV—a highly clinically relevant clinical question among medical specialists caring for patients in critical conditions.

As a future work, we will explore supervised topic models [14] to learn both the topics and predictions simultaneously. There are also more flexible neural network language models such as ClinicalBERT that can learn more abstract terms [15, 16]. We will compare our simpler topic model with ClinicalBERT. Moreover, we will also explore a powerful combination of recurrent neural network and topic model (TopicRNN) [17], which learns both the global context with the topic model and the local context with the RNN. Applications using an analogous idea of predicting readmission of ICU patients using billing code has also shown some promising results [18]. Lastly, our method is not limited to the healthcare domain. For example, we can model documents written in different languages or book reviews by literary scholars from different domains. Together, we envision that our current model can succeed in many application domains, where knowledge is manifested as free-form text in human natural language from diverse empirical domain-knowledge.

## Supporting information

**S1 File.**
(PDF)

## Author Contributions

**Conceptualization:** Naomi George, Charlotta Lindvall, Yue Li.

**Data curation:** Edward Moseley.

**Investigation:** Chih-Ying Deng, Naomi George, Charlotta Lindvall, Yue Li.

**Methodology:** Yue Li.

**Software:** Pratheeksha Nair, Yue Li.

**Supervision:** Charlotta Lindvall, Yue Li.

**Validation:** Zhi Wen, Pratheeksha Nair, Xing Han Lu.

**Writing – original draft:** Chih-Ying Deng, Edward Moseley, Naomi George, Charlotta Lindvall, Yue Li.

**Writing – review & editing:** Zhi Wen, Pratheeksha Nair, Chih-Ying Deng, Xing Han Lu, Edward Moseley, Naomi George, Charlotta Lindvall, Yue Li.

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
