## [Decision Letter · Decision Letter 0]

22 Sep 2020

PONE-D-20-09700

Mining heterogeneous clinical notes by multi-modal latent topic model

PLOS ONE

Dear Dr. Li,

Thank you for submitting your manuscript to PLOS ONE. After careful consideration, we feel that it has merit but does not fully meet PLOS ONE’s publication criteria as it currently stands. Therefore, we invite you to submit a revised version of the manuscript that addresses the points raised during the review process.

We look forward to receiving your revised manuscript.

Kind regards,

Ivan Olier, Ph.D.

Academic Editor

PLOS ONE

Journal Requirements:

2.Thank you for stating the following financial disclosure:

 [The funders had no role in study design, data collection and analysis, decision to publish, or preparation of the manuscript.].

Reviewers' comments:

Reviewer's Responses to Questions

**Comments to the Author**

1. Is the manuscript technically sound, and do the data support the conclusions?

Reviewer #1: Yes

Reviewer #2: Partly

Reviewer #3: Yes

2. Has the statistical analysis been performed appropriately and rigorously? 

Reviewer #1: Yes

Reviewer #2: No

Reviewer #3: Yes

3. Have the authors made all data underlying the findings in their manuscript fully available?

Reviewer #1: Yes

Reviewer #2: Yes

Reviewer #3: Yes

4. Is the manuscript presented in an intelligible fashion and written in standard English?

Reviewer #1: Yes

Reviewer #2: Yes

Reviewer #3: Yes

5. Review Comments to the Author

Reviewer #1: The authors have proposed an interesting study on the text modelling for electronic health records. In particular, the authors conceived that different note types should be modelled differently in a single framework. Therefore, they have proposed the modelling solution. Overall, the idea is well-motivated with good writing styles in a logical manner. In particular, I would like to highlight that the authors did ask for the physicians' involvements in the proposed study. Such experiments reflect the authors' efforts and its clinical relevance. I have the following minor comments:

1. It could be nice if the authors could show other methods for comparisons.

2. The time complexity analysis or running time could be provided.

3. The source code could be released.

Reviewer #2: # Review of Manuscript PONE-D-20-09700

## Mining heterogeneous clinical notes by multi-modal latent topic model

## Summary

I appreciated the opportunity to read this interesting paper. The authors propose an extension of the seminal Latent Dirichlet Allocation topic model of Blei et al. (2003) such that documents (i.e., medical notes) about a given patient written by different authors can be accommodated. This is accomplished by introducing multiple topic-word probability distribution matrices that are author-specific. They describe a variational Bayes estimation procedure to approximate the posterior distribution of the model parameters. They evaluated generalizability (i.e., prediction quality) using cross-validation on a subset of the MIMIC-III (Johnson et al., 2016). Results suggest marginal or no improvement in prediction quality over two alternative LDA models. Two facets of topic interpretability from the three models were evaluated by two external raters with relevant expertise. Overall, the proposed model is a useful and extensible generalization of LDA that could be useful in a variety of fields and applications beyond medicine.

## Major Issues

+ Page 3, 4: All notes for a given patient are assumed to have the same topic proportion vector $\\theta_j$. This seems to be a very strong and unrealistic assumption since each note may quite naturally contain different topics or different compositions of topics, a much more plausible scenario that the present model does not accommodate. For example, a nurse may write a note that focuses on a different subset of the K topics than a doctor.

+ Page 4: The notation in the variational algorithm equations is inconsistent and symbols and subscripts are not always clearly defined. Please carefully define the notation of the different counts $n$. In particular, Equation (5) introduces new notation that is never defined, specifically $\\Psi$ and $W_t$.

+ While an estimation algorithm for the proposed model is introduced, its performance is not evaluated. Without a simulation study to evaluate the quality of the point estimates obtained from the variational algorithm, it is impossible to know (a) if it can correctly recover the parameters of the proposed model (b) the necessary data requirements (e.g., number of documents, length of documents) and (c) the impact of potential complicating factors on model recovery (e.g., vocabulary size, the number of note types, impact of missing note types for some participants). Many applications of topic models in social science and medicine apply these models to much smaller data sets, particularly, short documents and a small number of documents where the models can break down. It would be valuable for both further methodological development and good practice in application to study the statistical performance of this model systematically.

+ Page 5, 6: What predictors were used in the logistic regression models? The topic proportion estimates $\\hat{\\theta}_j$?

+ Page 6: Why was the number of topics set to 50? Was this arbitrary or was this chosen by cross-validation on training data? The use of 50 topics could very likely contribute to overfitting especially given the large number of additional parameters being estimated in the proposed model. Since there is very little noticeable improvement in prediction quality, the data simply may not support a 50-topic model.

+ Page 6: The ROCs in Figure S2 do not support the claim that the "multi-note model achieved superior performance". The difference in AUROC is 86% vs. 85% and is presented without uncertainty estimates. Accounting for uncertainty, I would conjecture that the models are equivalent in prediction on this problem. This model does not, of course, have to be a major predictive breakthrough as its potential improvements to topic interpretability are interesting. However, I encourage the authors to (a) use less biased language when comparing the models' performance and (b) provide confidence intervals for the AUROC estimates throughout the paper.

+ Page 6, 7: Please define "cohesiveness". The claim that the "topics generated by the multi-note model had significantly more cohesiveness" is not supported unless I am missing something. This appears to be an entirely subjective claim.

+ Page 9: The claims in the last two sentences are overly strong given the evidence provided. It is not clear that the proposed model necessarily provides more meaningful topics (see comments elsewhere on this point). There is certainly limited or no evidence on this data set that the proposed model provides more accurate prediction of MV duration. As noted elsewhere, the differences in AUROC between the different models is negligible. Of course, this model *may* provide better predictive performance on other data sets. Indeed, as the authors hint at on p. 10, because the topics are not linked in the model to the outcome of interest as in, for example, supervised topic models (Blei & McAuliffe, 2008), there is no reason for the topic proportion estimates to be related at all to the outcome.

+ Please provide code to reproduce the analyses described throughout the paper.

## Minor Issues

+ Figure 1, p. 3: $z_{ijk}$ and $x_{ij}$ should have superscripts $(t)$ to differentiate the different notes for patient $j$ These superscripts appear in the distributional specification of both random variables on page 3.

+ Page 3, 4: References to the multinomial distribution should be replaced with the **categorical** distribution. This is also consistent with the distribution definitions for $z_{ij}^(t)$ and $x_{ij}^(t)$, which lack the normalization constants of a multinomial distribution but are consistent with the categorical distribution.

+ Page 5. Summary statistics of the document/note text should be provided. How long are these notes? How large is the vocabulary of each type of note?

+ Page 5. Was the impact of the pre-processing choices on model performance evaluated at all?

+ Page 5, 6: The outcome being predicted in the empirical application was dichotomized from a count variable (number of days), which can often reduce the information available and is not generally recommended. Please justify this decision or, perhaps better, use the original outcome.

+ Page 6 and elsewhere: Language used to compare the proposed model to the comparison models is overly optimistic and potentially misleading. A difference in AUROC of 66% vs. 65% is very small. Indeed, confidence intervals for the AUROC should be provided for all models as I suspect that they will indicate substantial uncertainty.

+ Page 7: What do the numbers in parentheses for the average interpretability scores represent? Standard deviations? Confidence limits? On a related note, if the ratings are between 1 and 5, a rating of 4.22 + 1.15 = 5.37 is impossible. Finally, a direct test comparing the interpretability rating of the rater(s) would be useful to avoid overinterpretting what appear to be quite noisy estimates of interpretability. Accounting for uncertainty, there may be no statistically significant difference in interpretability between the two models.

+ Page 8: What measure of correlation was used? Pearson's correlation (the usual default) is inappropriate for correlating mixture proportion estimates with a duration variable because of the range restriction on the mixture proportion estimates. A better option would be Spearman's or Kendall's correlation coefficients.

+ Figure 4, Page 9: Please descibe the components of this boxplot as different graphing software uses different conventions. Are the whiskers quartiles? Confidence limits? What do the bounds of the box indicate? What are the points indicating and are they overplotted (jitter the points if so)?

## Typos

+ Page 4: After Equation (3), please remove the fragment "The hyperparameters $\\alpha_k$ and $\\beta_{tk}$".

Reviewer #3: The authors introduced a novel hierarchical topic model method for clinical notes retrieved from Harvard teaching hospitals. The method and its interpretation are interesting from both machine learning/NLP and clinical informatic perspective. I have following comments

1. The topic modelling method, by its nature, belongs to unsupervised distributional embedding learning, which is a traditional branch of text embedding widely used. In recently years, the clinical NLP community has been dominant by “BERTologists” ( the group of researchers focusing on transformer based model). Could the authors clarify the different use cases and advantages of method present in this manuscript compared with transformer based models?

2. One interesting feature found in this method is topic discovery and interpretability of the model. I would suggest the author reflect these points in abstract and even the tittle for readers without much machine learning backgrounds.

3. When predicting mortality, models present in this manuscript has AUCROC<0.70. As MIMIC3 dataset is from ICU and many publications claim AUCROC >0.80, it is necessary for the authors to justify this relatively low performance. If the focus is not performance, it should be clearly mentioned early in the text.

6. PLOS authors have the option to publish the peer review history of their article (what does this mean?). If published, this will include your full peer review and any attached files.

Reviewer #1: No

Reviewer #2: No

Reviewer #3: No

---

## [Author Response · Author response to Decision Letter 0]

3 Dec 2020

Please see Response to Reviewers.pdf

---

## [Decision Letter · Decision Letter 1]

20 Jan 2021

PONE-D-20-09700R1

Mining heterogeneous clinical notes by multi-modal latent topic model

PLOS ONE

Dear Dr. Li,

Thank you for submitting your manuscript to PLOS ONE. After careful consideration, we feel that it has merit but does not fully meet PLOS ONE’s publication criteria as it currently stands. Therefore, we invite you to submit a revised version of the manuscript that addresses the points raised during the review process.

We look forward to receiving your revised manuscript.

Kind regards,

Ivan Olier, Ph.D.

Academic Editor

PLOS ONE

Reviewers' comments:

Reviewer's Responses to Questions

**Comments to the Author**

1. If the authors have adequately addressed your comments raised in a previous round of review and you feel that this manuscript is now acceptable for publication, you may indicate that here to bypass the “Comments to the Author” section, enter your conflict of interest statement in the “Confidential to Editor” section, and submit your "Accept" recommendation.

Reviewer #1: All comments have been addressed

Reviewer #2: (No Response)

2. Is the manuscript technically sound, and do the data support the conclusions?

Reviewer #1: Yes

Reviewer #2: Partly

3. Has the statistical analysis been performed appropriately and rigorously? 

Reviewer #1: Yes

Reviewer #2: No

4. Have the authors made all data underlying the findings in their manuscript fully available?

Reviewer #1: Yes

Reviewer #2: Yes

5. Is the manuscript presented in an intelligible fashion and written in standard English?

Reviewer #1: Yes

Reviewer #2: No

6. Review Comments to the Author

Reviewer #1: The authors have addressed my comments satisfactorily. In particular, the authors have collaborated with the medical side which is very nice.

Reviewer #2: ## Summary

I thank the authors for addressing most of the questions and comments I had from the first version of their manuscript. In particular, I applaud them for conducting and reporting a simulation study to evaluate the performance of their estimation algorithm. This provides valuable information about the proposed algorithm's ability to successfully estimate the proposed model, particularly for data sets with similar characteristics. I also thank the authors for providing access to the code for the proposed algorithm and its application via Github. I have several remaining points for the authors to consider, which I list below.

## Interpretability

+ Page 7, Line 159-160: As I mentioned in my review of the original manuscript, a direct test comparing the interpretability rating of the rater(s) would be useful to directly test the difference in interpretability ratings between the topics from the proposed model and the single-note model. I did so using paired t-tests (under a range of assumed correlations since I don't have the original scores) and find that the difference is statistically significant across different potential correlations, so I agree that there is improvement in physician rating of interpretability. My main suggestion here is to report such a test directly in addition to the means and standard deviations to provide clear evidence of the claim that is highlighted in both the abstract and in the Discussion (Line 190-193) that interpretability may be better using the multimodal topic model instead of the standard LDA model in this example.

## Prediction

+ Page 5, Line 131-132: As I mentioned in my review of the original manuscript, the ROCs in Figure 2 and in Figure S2 do not support the claim here that the "multi-note model performed better compared to single-note models". For example, the difference in AUROC from Figure S2 when predicting whether a patient will be on a mechanical ventilator for more than 7 days or not is 86.0% vs. 85.2%, an arguably negligible difference, which is presented without uncertainty estimates such as the standard deviation across the 5-fold cross-validation. Accounting for uncertainty, I would conjecture that the models are equivalent in prediction on this problem. Again, I encourage the authors to provide confidence intervals for the AUROC estimates throughout the paper (e.g., via bootstrapping as in the pROC R software package; Robin et al., 2011, doi: 10.1186/1471-2105-12-77).

+ As I mentioned in my original review and in the first bullet point above, claims in the results sections and discussion (e.g., Lines 130-132, 193-194) are not supported by the evidence provided: The authors claim that the proposed model "performed better compared to single-note models, as measured by AUROC and AUPRC" and "our model confers more accurate prediction of duration of MV". However, the AUROC and AUPRC values are, as I mentioned above, quite similar. There is no strong statistical evidence (unless confidence intervals for each AUROC/AUPRC value per model and for the difference in AUROC/AUPRC between models are provided) that prediction performance differed between the two models. I would echo comments from other reviewers that even if these AUROC values differ significantly from a purely statistical inference perspecive, that the very small magnitude of such a difference needs to be defended. Alternatively, the authors could omit claims that their model's predictive performance is better than single-note models and focus on the improved interpretability of the topics extracted by their model. To clarify, I do not believe that they need to show better predictive performance for this paper to be a valuable contribution to the literature. However, I do not want their claims in the results and discussion to potentially misread a casual reader.

## Errata

While the overall conclusions and results in the paper are clear, the paper would benefit from careful proofreading for minor typographical and grammatical errors (e.g., definite and indefinite articles such as "a" and "the" are omitted throughout the paper) before publication.

7. PLOS authors have the option to publish the peer review history of their article (what does this mean?). If published, this will include your full peer review and any attached files.

Reviewer #1: No

Reviewer #2: No

---

## [Author Response · Author response to Decision Letter 1]

20 Feb 2021

Please see Response to Reviewers.pdf

---

## [Decision Letter · Decision Letter 2]

23 Mar 2021

Mining heterogeneous clinical notes by multi-modal latent topic model

PONE-D-20-09700R2

Dear Dr. Li,

We’re pleased to inform you that your manuscript has been judged scientifically suitable for publication and will be formally accepted for publication once it meets all outstanding technical requirements.

Kind regards,

Ivan Olier, Ph.D.

Academic Editor

PLOS ONE

Additional Editor Comments (optional):

Reviewers' comments:

Reviewer's Responses to Questions

**Comments to the Author**

1. If the authors have adequately addressed your comments raised in a previous round of review and you feel that this manuscript is now acceptable for publication, you may indicate that here to bypass the “Comments to the Author” section, enter your conflict of interest statement in the “Confidential to Editor” section, and submit your "Accept" recommendation.

Reviewer #1: All comments have been addressed

Reviewer #2: All comments have been addressed

2. Is the manuscript technically sound, and do the data support the conclusions?

Reviewer #1: Yes

Reviewer #2: Yes

3. Has the statistical analysis been performed appropriately and rigorously? 

Reviewer #1: Yes

Reviewer #2: Yes

4. Have the authors made all data underlying the findings in their manuscript fully available?

Reviewer #1: Yes

Reviewer #2: Yes

5. Is the manuscript presented in an intelligible fashion and written in standard English?

Reviewer #1: Yes

Reviewer #2: Yes

6. Review Comments to the Author

Reviewer #1: The authors have addressed my comments already.I don't have any further commet. Thank you very much.

Reviewer #2: (No Response)

7. PLOS authors have the option to publish the peer review history of their article (what does this mean?). If published, this will include your full peer review and any attached files.

Reviewer #1: No

Reviewer #2: No

---

## [Editor Report · Acceptance letter]

29 Mar 2021

PONE-D-20-09700R2 

Mining heterogeneous clinical notes by multi-modal latent topic model 

Dear Dr. Li:

I'm pleased to inform you that your manuscript has been deemed suitable for publication in PLOS ONE. Congratulations! Your manuscript is now with our production department. 

Kind regards, 

on behalf of

Dr. Ivan Olier 

Academic Editor

PLOS ONE